# A Comparison Study on Polysaccharides Extracted from *Atractylodes chinensis* (DC.) Koidz. Using Different Methods: Structural Characterization and Anti-SGC-7901 Effect of Combination with Apatinib

**DOI:** 10.3390/molecules27154727

**Published:** 2022-07-24

**Authors:** Pingfan Zhou, Wanwan Xiao, Xiaoshuang Wang, Yayun Wu, Ruizhi Zhao, Yan Wang

**Affiliations:** 1School of Chinese Materia Medica, Guangdong Pharmaceutical University, Guangzhou 510006, China; pinkgiee@163.com (P.Z.); w989700@163.com (W.X.); 2The Second Affiliated Hospital of Guangzhou University of Chinese Medicine, Guangzhou 510006, China; m18314484252@163.com (X.W.); wyy_1220@gzucm.edu.cn (Y.W.); 3Guangdong Provincial Key Laboratory of Clinical Research on Traditional Chinese Medicine Syndrome, Guangzhou 510006, China

**Keywords:** *Atractylodes chinensis* (DC.) Koidz., different extraction methods, polysaccharide, chemical structure, synergistic activity

## Abstract

For hundreds of years, *Atractylodes chinensis* (DC.) Koidz. (AK) has been widely used as a treatment for spleen and stomach diseases in China. The AK polysaccharides (AKPs) have been thought to be the important bioactive components. In this stud, the impacts of different extraction methods were analyzed. The differences between AKPs extracted by hot water extraction (HWE), AKPs extracted by ultrasonic extraction (UAE), and AKPs extracted by enzyme extraction (EAE) were compared in terms of yield, total carbohydrate content, molecular weight distribution, monosaccharide composition, and synergistic activity of the AKPs with apatinib were determined. The results indicated that the yield of the polysaccharide obtained from HWE was higher than that of UAE and EAE. However, activity assays indicated that UAE-AKPs and HWE-AKPs enhanced apoptosis of human gastric cancer cells (SGC-7901) treated with apatinib and UAE-AKPs showed the strongest synergistic activities. This is also in agreement with the fact that UAE-AKPs have a smaller molecular weight, β-configuration, and higher galactose content. These findings suggested that UAE is an efficient and environmentally friendly method for producing new polysaccharides from *Atractylodes chinensis* (DC.) Koidz. for the development of natural synergist and for the treatment of gastric cancer.

## 1. Introduction

Gastric cancer (GC) is the fifth most common tumor and the third most deadly malignancy. It possess a higher mortality/incidence ratio (0.845) and a five-year prevalence (27.6/100,000) in China, which is significantly more than in other developed countries [1]. Unfortunately, the current diagnosis of gastric cancer is mainly advanced, and patients must rely on traditional drugs such as apatinib (apatinib mesylate) which is used to keep them alive [2,3]. However, due to the large number of toxic side effects of apatinib, patient treatment is not ideal, and clinical applications of apatinib are limited [4,5]. Fortunately, combination therapy is an effective way to solve this dilemma [6]. Therefore, there is an urgent need to find new synergistic drugs to effectively combine with the drugs. Studies have shown that many meridian-guiding Chinese medicines and their main ingredients, such as polysaccharide, have synergistic effects on drugs [7,8,9].

Atractylodes chinensis (DC.) Koidz. (AK) is a meridian guide drug which belongs to the stomach and spleen meridian [10,11]. Its rhizomes have been used as traditional Chinese medicine (TCM) to treat diseases of digestive disorders [12], which are listed as stomachic drugs in Korean and Japanese pharmacopeias [13]. Its high medicinal value is related to the physiologically active substance *Atractylodes chinensis* (DC.) Koidz. polysaccharides (AKPs) [14]. AKPs, as with most polysaccharides, are macro molecular carbohydrates composed of several monosaccharide units which are involved in various biological activities, such as enhancing immunity, inhibiting tumor cell growth, and inducing apoptosis [15,16,17]. At the same time, many studies have found that natural polysaccharides derived from TCM have a significant synergistic activity due to their immune enhancement of the body or direct cytotoxicity to cancer cells [18]. However, until now, the synergistic activity of AKPs has not been thoroughly investigated.

Furthermore, extraction techniques greatly affect the biological activity of polysaccharides [19,20]. Therefore, it is very important to select a suitable extraction method to protect and improve the biological activity of polysaccharides. In recent years, various extraction techniques for the extraction of AKPs have been developed, mainly, including hot water extraction (HWE), ultrasonic extraction (UAE), and enzymatic extraction (EAE). However, previous studies on AKPs extraction have mainly focused on the yield or the effect of a single variable and have not investigated the possibility of obtaining a comprehensive evaluation of the yield and activity of polysaccharides [21]. In addition, the physicochemical properties of polysaccharides, such as chemical composition, molecular weight distribution, and type of glycosidic bond, were significantly influenced by different extraction processes [22]. For example, Chen et al. [23] evaluated the effect of four extraction methods on the characteristic and antioxidant activities of the polysaccharides of the comfrey polysaccharides, and the results indicated that the higher content of furoic acid and smaller molecular weight of the polysaccharide of the ultrasonic-assisted method might have higher antioxidant activities. To the best of our knowledge, information on the effects of ultrasonic extraction or/and enzymatic extraction on the physicochemical properties and bioactivities of AKPs has not yet been established.

Therefore, in this study, three techniques, namely HWE, UAE, and EUA, were employed to extract AKPs and their physicochemical characteristics such as chemical composition, molecular weight and monosaccharide composition were studied. Synergistic activities were used to evaluate that AKPs with apatinib inhibited SGC-7901. More importantly, the study was designed to provide deeper insight and develop promising applications for AKPs as an ingredient of medicinal value.

## 2. Materials and Methods

### 2.1. Material and Chemical Reagents

*Atractylodes chinensis* (DC.) Koidz. (Lot: 201201) was purchased from Guangzhou Zisun Pharmaceutical Co., Ltd. (Guangzhou, China), authenticated by Nengfeng Ou, the pharmacist in charge of herb quality in the Second Affiliated Hospital of Guangzhou University of Chinese Medicine. Cellulase (Lot: L03J11L117193) and pectinase (Lot: SLGD2368) were obtained from Shanghai Yuanye Bio-Technology Co., Ltd. (Guangzhou, China) and SIGMA Co., Ltd. (St. Louis, MO, USA) Shanghai Macklin Biochemical Co. Ltd. supplied Apatinib Mesylate (YN968D1, Lot: C12457576) (Shanghai, China). GIBCO provided the RPMI-1640 medium (Lot: 8119141), Fetal bovine serum (FBS) (Lot: 2176398) and Penicillin Streptomycin (Lot: 2199829) (Gaithersburg, MD, USA). MP Biomedicals, LLC provided the 3-(4,5-dimethylthiazol-2-yl)-2,5-diphenyl-2H-tetrazolium bromide (MTT) (Lot: 7101KMP) (Santa Ana, CA, USA,). MultiSciences provided the cell cycle kits (Lot: A01124) (Hangzhou, China). Human gastric cancer cell line SGC-7901 were acquired from Shanghai gefan biotechnology Co. Ltd. (Shanghai, China). American Polymers Standards Corporation supplied Dextran standards (Dextran 3690K (Lot: D620D10-1), Dextran 2370K (Lot: D620D7-1), Dextran 680K (Lot: D620D8-2), Dextran 450K (Lot: D620DC-3), Dextran 60K (Lot: D620D1-9) and Dextran 4K (Lot: 8842) (Cleveland, OH, USA). Yuanye Biotechnology Co., Ltd. standards (d-mannose (Lot: 3458-28-4), d-galactose (Lot: Z22J9H64187), arabinose (Lot: 10323-20-3), galacturonic acid (Lot: 91510-62-2), glucose (Lot: 50-99-7) and rhamnose (C11366744) (Shanghai, China). 

### 2.2. Preparation of AKPs Extracted by Different Methods

#### 2.2.1. Pretreatment of AK

Firstly, *Atractylodes chinensis* (DC.) Koidz. (AK) pieces were ground for 3 min in a vibrating cellular ultrafine crusher (800C, Xiantao Co., Ltd.) and sieved with 50 mesh sieves. The AK powder was degreased with ethanol (5:1 = *v*/*w*, 70%, *v*/*v*) for 2 h at 80 °C to remove ethanol-soluble constituents, then repeated four times. The liquid was discarded after filtration, and the residue was dried at 40 °C.

#### 2.2.2. Extraction of AKPs

AK material (100 g) was mixed with distilled water (2000 mL; material-to-liquid ratio 1:20) and extracted according to the best parameters of three different techniques (HWE, UAE, EAE) as previously described [24,25,26] with some modifications. The polysaccharides thus obtained were named HWE-AKPs, UAE-AKPs and EAE-AKPs respectively. The extraction of HEW-AKPs was carried out by reflux extraction in a water-bath at 90 °C for 4 h; it was repeated twice. The extraction of UAE-AKPs was carried out in an ultrasonic bath (TLCX-600, Sky Blue Biotechnology Co., Ltd., Tianjing, China) with a power of 150 W at 60 °C for 40 min, repeated twice. The extraction of EAE-AKPs polysaccharides was performed using a complex enzyme, which was composed of pectinase and cellulase in a 2:1 ratio. The fraction of complex enzyme in the extraction solvent was 1.2%. Enzymatic extraction was carried out at 53 °C for 90 min on a shaker with constant temperature (MaxQ&8000, Thermo Scientific^TM^, Waltham, MA, USA). Subsequently, the temperature was rapidly increased to 80 °C for 1 h to deactivate the enzyme.

After extraction, the solutions were centrifuged at 3000 rpm for 15 min at room temperature. The supernatant was concentrated to 0.5 mg crude drug per mL using a rotary evaporator at 50 °C under a vacuum. After being cooled to room temperature, the solution was precipitated overnight at 4 °C, with ethanol (95%) to obtain a final concentration of 80% (*v*/*v*). The precipitate was collected and washed with anhydrous ethanol, redissolved in distilled water, and lyophilized in a vacuum freeze dryer (Labconco FreeZone ^®^^6^, Labconco Co., Kansas City, MO, USA) to obtain crude AKPs. The crude AKPs were subjected to the Sevage method (4:1 chloroform: butyl alcohol) to remove free proteins. The deproteinated solution was dialyzed against distilled water for 72 h (MWCO 400 Da). Finally, the retentate in the dialysis bag was collected, concentrated, and lyophilized in a vacuum freeze dryer (Labconco FreeZone ^®6^, Labconco Co., Kansas City, MO, USA). The polysaccharide yield was calculated using the following formula:(1)$$\mathrm{Polysaccharide}\;\mathrm{yield}\;\left(\%,\frac{w}{w}\right)=\frac{\mathrm{weight}\;\mathrm{of}\;\mathrm{dried}\;\mathrm{AKPs}\;\left(g\right)}{\mathrm{weight}\;\mathrm{of}\;\mathrm{pretreated}\;\mathrm{AKs}\;\mathrm{powdera}\;\left(g\right)}\;\times \;100$$

### 2.3. Characterization of Different AKPs

#### 2.3.1. Analysis of Ultraviolet-Visible (UV–Vis) Spectroscopy

Ultraviolet-visible (UV) spectroscopy was used as per the method reported, with minor modifications [27]. Briefly, three 1.5 mg AKPs (UAE-AKPs, HWE-AKPs, EAE-AKPs) dissolved in 1 mL of water at a concentration of 1.5 mg/mL was scanned by the U-2910 ultraviolet spectrophotometer (Hitachi Hi-Tech Co., Tokyo, Japan) from the wavelength range of 600 nm to 190 nm, respectively. 

#### 2.3.2. Analysis of Fourier Transform-Infrared (FT-IR) Spectroscopy

The dried AKPs samples (2 mg) were mixed with spectroscopic KBr powders (300 mg), ground and pressed into pellets. The sample spectrum was recorded in the range of 400–4000 cm^−^^1^ with an FT-IR spectrophotometer (Spectrum Two, PerkinElmer, Waltham, MA, USA) as previously described in Minjares-Fuentes et al. [28].

#### 2.3.3. Determination of the Total Carbohydrate Content of AKPs 

The total carbohydrate content was measured according to the phenol-sulfuric acid method [29] using d-glucose as the standard. The glucose standard solution with a concentration of 500 µg/mL was prepared by 1.01 mg d-glucose in 2 mL of ultrapure water and sequentially half-diluted to 250 µg/mL, 125 µg/mL, 62.5 µg/mL, 31.25 µg/mL, 15.63 µg/mL, 7.81, and 3.90 µg/mL. A 100 µL of 5% phenol was added to the 200 µL of the different concentrations of standard glucose solutions prepared. Subsequently, sulfuric acid was rapidly added until the total volume was 800 µL. The samples were kept at room temperature for 15 min and then thoroughly shaken. Subsequently, 150 µL of each sample was loaded into the microplate and measured with the microplate reader at 490 nm (Biotek, Synergy H1, Winooski, VT, USA). The standard curve was obtained with the absorbance as the coordinate (y) and the glucose concentration as the abscissa (x). The AKPs prepared as a 0.5 mg/mL sample solution were also determined according to the same procedures as above. Finally, the total sugar content of the samples was determined using a linear equation.

#### 2.3.4. Determination of the Molecular Weight Distribution of AKPs

The average molecular weight (Mw) was determined using high-performance gel permeation chromatography (HPGPC) as described by Liu et al. [8]. A solution of 1 mg of AKPs was made by dissolving in 2.0 mL of water and filtered through a 0.45 µm pore size membrane filter. Chromatographic conditions were as follows: Agilent 1260 HPLC system (Agilent Technologies, Santa Clara, CA, USA) and TSK gel G5000PWXL column (7.8 mm × 300 mm, Tosoh Corporation, Yamaguchi, Japan); data processing system: Agilent GPC software, mobile phase ultrapure water, flow rate was 0.4 mL/min, column temperature was set at 40 °C, and injection volume was 10 µL. Dextran T series standards were used to create a standard curve (standard molecular weights of 3690, 2370, 1190, 680, 550, 450, 220, 90, 60, 35, and 4 kDa, respectively).

#### 2.3.5. Monosaccharide Composition of AKPs

The monosaccharide composition was determined as reported by Chen et al. [30]. High-performance liquid chromatography (HPLC) and PMP derivative were used to measure the monosaccharide composition of AKPs.

AKPs, 5 mg were precisely weighed and hydrolyzed with 5 mL of 4 mol/L trifluoroacetic acid (TFA) at 110 °C for 6 h in a sealed tube. The excess acid was evaporated with a rotary evaporator after the hydrolysis was complete. Subsequently, 5 mL of methanol was added and dried using the same method described above, and the procedure was repeated three times for the removal of TFA [31]. The hydrolysates were then cooled to room temperature, mixed with 1-Phenyl-3-Methyl-5-Pyrazolone (PMP) (300 µL), 0.3 mol/L NaOH (300 µL), and stored at 70 °C for 100 min. The reactants were again cooled to room temperature, and 0.3 mol/L of HCl (300 µL) was added to neutralize the reaction; after; 1 mL of CHCl_3_ was used to extract and collect the supernatant. Finally, the solution was passed through a 0.45 µm syringe filter for HPLC analysis. Meanwhile, monosaccharide standards, including d-mannose (Man), d-galactose (Gal), arabinose (Ara), galacturonic acid (GalA), glucose (Glc), rhamnose (Rha), were prepared as described above. 

Subsequently, the constituent monosaccharides of each sample were determined using an HPLC system (Agilent 1200, Agilent Technologies Co. Ltd., Santa Clara, CA, USA) equipped with a Diamonsil C_18_ column (250 × 4.6 mm, 5 µm) and a diode array detector (DAD, Agilent Technologies, Santa Clara, CA, USA). A 10 µL sample was injected into the HPLC system at 40 °C and eluted with a mixture of 0.05 mol/L phosphate (Na_2_HPO_4_–NaH_2_PO_4_, pH = 6.83) and acetonitrile (83:17, *v*/*v*) at a flow rate of 1.0 mL/min. The DAD wavelength was set to 254 nm. 

### 2.4. In Vitro Synergistic Activity Study

#### 2.4.1. Cell Culture

SGC-7901 human gastric cancer cell lines were maintained in RPMI-1640 medium. The medium was supplemented with 10% heat-inactivated FBS (*v*/*v*) and 1% penicillin/streptomycin in a humidified atmosphere containing 5% CO_2_.

#### 2.4.2. MTT Assay for Cytotoxicity

##### Cell Cytotoxicity Assay

The cell proliferation of apatinib and AKPs against human gastric carcinoma SGC-7901 was determined using the MTT assay as described by Wu et al. [32]. Cells were seeded in 96-well plates at a density of 1.5 × 10^4^ cells/mL (200 µL) and incubated for 24 h. The various concentrations of apatinib (5, 10, 20, 40, 80 µmol/L) were added to tumor cells, respectively, and AKPs (1600, 800, 400, 200,100, 50 µg/mL) were also added. The same volume of medium containing 0.5% dimethyl sulfoxide (DMSO) was added and treated as blank control. After 72 h of incubation, each well was added with 20 µL MTT and incubated for another 4 h to protect from light. Subsequently, the supernatant was removed, and 150 µL of DMSO was added to each well, vortexed for 15 min in the absence of light. The optical density (*OD*) values were measured at 490 nm by using a microplate reader (Bio-Tek, EXl800, Winooski, VT, USA). Cell viability was calculated as using the following formula:(2)Inhibitory rate%=(ODcontrolODexperimental grop)ODcontrol×100%

##### Analysis of the Synergistic Effect of the Combination Apatinib with AKPs

Each set-up group set up contained 4 wells. Apatinib (30 µmol/L), AKPs (0.5, 0.25, and 0.125 µg/mL), or a combination of both was added to cells in the logarithmic phase; after 72 h of drug action, the MTT method was used to detect cell viability. The combination index was expressed in the calculation of the Q value [9]. Synergism: Q > 1.15; additive effect: Q = 0.85–1.15; antagonism: <0.85.
(3)$$Q=\frac{E_{AB}}{\left(E_{A}+E_{B}-E_{A}\times E_{B}\right)}$$

#### 2.4.3. Cell Cycle Analysis

Cell cycle were analyzed by flow cytometry using Annexin V-FITC/PI cell cycle staining kit. Staining was performed according to the instructions provided by the manufacturer, and flow cytometry was performed (NovoCyte Quanteon, Agilent Technologies Co. Ltd., Santa Clara, CA, USA). SGC-7901 cells were grown in 12-well plates with culture under standard growth conditions until they reached 70% confluency. Then, these were exposed to various treatments for 72 h. Based on the results of the MTT experiment, 0.5 µg/mL AKPs were combine with apatinib (30 µmol/L) as a treatment group. The same volume of medium was used as a blank.

### 2.5. Statistical Analysis 

All experiments were performed in triplicate and data were expressed as means ± standard deviation (SD). Origin 2021 (OriginLab Corporation, Northampton, NC, USA) and GraphPad Prism 8 (GraphPad Software Inc., San Diego, CA, USA) were used for drawing processing and statistical analysis of data. Statistical significances were carried out by one-way analysis of variance (ANOVA). Values of *p* < 0.05 were considered as statistically significant. 

## 3. Results

### 3.1. Yield and Total Carbohydrate Content of AKPs 

As shown in Table 1, the yields of the AKPs were significantly influenced by different extraction techniques, and this was in agreement with the results of Chen [33]. The yield of HEW-AKPs (13.44%) was significantly higher than that of UAE-AKPs (3.27%) and EAE-AKPs (2.125%), suggesting that the higher yield could be due to the higher temperature and the longer extraction time of extraction, which was consistent with the study of Hammi et al. [34]. Furthermore, more than 110 W of extracted power reportedly produced an excessive bubbling, which reduced the efficiency of ultrasound energy transmitted into the medium [35], thus decreasing the yield. Furthermore, UAE and EAE have a lower yield than HWE, which can be caused by ultrasound and enzyme degradation, inducing fragmentation of sugar chains and polysaccharides [36] Furthermore, a loss of yield was observed, which could be due to the precipitation of alcohol [37]. Furthermore, according to the phenol-sulfuric, we also analyzed the carbohydrate content of each product. The results indicated that the different extraction methods did not have significant effects on the carbohydrate content (96.83–91.12%).

### 3.2. Ultraviolet-visible (UV–Vis) Spectroscopy

The UV-Vis profiles of AKPs were presented in Figure 1A. No obvious peaks were observed in AKPs, suggesting that there were trace amounts or no nucleic acid and protein [38]. 

### 3.3. Fourier Transform Infrared Spectroscopy (FI-IR)

FT-IR spectra recorded in the region of 4000–500 cm^−1^ for AKPs fractions were shown in Figure 1B. Most of the functional groups in the AKPs were assigned as reported in the literature [39]. In our results, all of the three fractions exhibited a broad intense peak near 3500–3100 cm^−1^, which represents intramolecular and intermolecular O–H stretching vibration [40]; a weak peak near 3000–2900 cm^−1^ was for the C–H stretching vibration and the symmetric deformation vibration, respectively [41]. These were the characteristic peaks of the polysaccharides. The absorption peak at 1740 cm^−1^ of the AKPs were symmetrical vibration peaks of carboxyl groups (C=O symmetrical stretching) [42]. Meanwhile, the existence of C=O is indicative of polysaccharides containing carboxylic acid or glyoxylates. Previous studies have demonstrated that there are strong absorption peaks at 1075–1010 cm^−1^ due to the stretching vibrations of pyranose [43,44], and this suggested that the sugars were in the pyranose form. Furthermore, when pyranose ring are linked, the absorption peaks at 875.4 cm^−1^ are generally expressed as beta-glycoside bonds, and at 815.7 cm^−1^, the observed peaks are alpha-glycoside bonds [45,46]. HWE and EAE coincided with alpha-pyranose, while UAE was beta-pyranose. This suggests that ultrasound could break the glycoside bonds of polysaccharides and change the glycoside bond configuration of polysaccharides [47,48].

### 3.4. Determination of the Molecular Weight Distribution of AKPs

HPGPC was used to determine the average molecular weight (Mw) of all AKPs. The standard curve was constructed using dextran standards of varying molecular weight: y = −0.1972x + 6.2045 (R^2^ = 0.9985). Figure 2 show the retention time and molecular weight obtained from various samples. Both UAE-AKPs and HWE-AKPs have two major peaks, high molecular peak I and low molecular peak II. The ratio of peaks I for UAE-AKPs and HWE-AKPs was 40.24% and 55.40%, respectively; in peak I and peak II, HWE-AKPs (3649.36 kDa) has a larger molecular weight than that of UAE-AKPs (3203.47 kDa). For EAE-AKPs, sharp peak I was not observed. EAE-AKPs has a wide molecular weight (5526.6~4159.2 kDa (28.39%)) and contains a large number of medium molecular weight polysaccharides (1734~390 kDa (18.91%)) compared to the other two. For three AKPs, the peak II is 10 kDa, but the ratio was different. The ratio in a decreased order indicated UAE-AKPs (52.05%) was the largest, followed by EAE-AKPs (47.33%) and HWE-AKPs (37.30%). This indicated that the molecular weight distribution of AKPs showed great variation; UAE and EAE techniques could obtain AKPs with lower molecular weights that may be related to the effect of acoustic cavitation [49] and cellulase treatment. In addition, both showed degradation of sugar chains [50], which was similar to the results reported by Chen et al. [51].

### 3.5. Monosaccharide Composition of AKPs

As shown in Table 2 and Figure 3, the monosaccharide composition of the three AKPs was identified based on the retention time of the reference monosaccharide. The standard curve was as follows: Man: y = 18.291x + 174.99 R^2^ = 0.9979; GlcA: y = 19.031x + 130.38 R^2^ = 0.9987; Rha: y = 33.235x + 311.19 R^2^ = 0.9991; GalA: y = 16.329x + 188.72 R^2^ = 1; Glu: y = 19.326x + 222 R^2^ = 0.9985; Gal: y = 20.311x + 111.9 R^2^ = 0.9984 Ara: y = 24.948x + 152.35 R^2^ = 0.9985. All AKPs exhibited the same types of monosaccharide, including Man, GlcA, Rha, GalA, Glu, Gal, and Ara, in varying proportions. Ara was found to be the predominant monosaccharide (64.02–54.65%) in the three AKPs. Additionally, the proportions of Gal, Gala and Glc were significantly different in the three AKPs. The HWE-AKPs had the highest Gluratio, whereas the EAE-AKPs contained the highest GalAratio. The ratio of Galain AKPs decreased in the following order: UAE-AKPs (22.80%) > EAE-AKPs (12.65%) > HWE-AKPs (7.54%).

### 3.6. In Vitro Synergistic Activity Study

#### 3.6.1. Apatinib or AKPs Inhibited Cell Proliferation

The results of the MTT assay showed that apatinib inhibited the proliferation of SGC-7901 cells in a dose dependent manner; the IC_50_ was 26.39 µmol/mL. The effect of the three AKPs on SGC-7901 cells was different. HWE showed a trace effect on growth promotion, while UAE-AKPs and EAE-AKPs effect did not have an on SGC-7901 proliferation.

#### 3.6.2. AKPs Promoted Apatinib against Cell Proliferation

Subsequently, the synergy effect was studied. The inhibition rates and combined indices (Q) are shown in Figure 4A,B. The results showed that after 72 h of treatment, with respect to apatinib, medium and high dose of UAE-AKPs and a medium dose of HWE-AKPs enhanced the effect of apatinib significantly, the inhibition rate increased to 25.22%, 26.57% and 16.07%, respectively. The synergy index was 2.18, 2.28 and 1.17. The effect of UAE-AKPs was dose-dependent. However, compared to apatinib, cell vitality changed marginally when combined with all three doses of EAE-AKPs.

#### 3.6.3. Effects of AKPs Combined with Apatinib on Cell Cycle

The cell cycle was investigated since it is closely related to proliferation. After 72 h of administration, cells of different groups were treated with a Cell Cycle Staining Ki cycle kit and detected with a Cytometer (NovoCyte Quanteon, Agilent Technologies Co. Ltd., Santa Clara, CA, USA). The results were presented in Figure 5, G1/G0 of the blank group was 41.79 ± 2.99%. Apatinib blocked SGC-7901 cells from gastric cancer in the G1/G0 stage; hence, the G1/G0 was 55.23 ± 0.81%. Compared with apatinib, the UAE-AKPs combination group improved the effect of apatinib cell cycle arrest effect of apatinib (G1/G0 64.58 ± 1.77%), but the other two AKPs had no effect on enhancing the arrest of the cell cycle of apatinib. This was consistent with the MTT results.

## 4. Discussion

Recently, natural polysaccharides have attracted a great deal of attention due to their advantages in synergistic and detoxifying effects. In addition, their potential biological activities, such as, immunomodulatory, antioxidant, antitumor, anti-inflammatory, antidiabetic, and antiviral activities, are being realized [52,53]. However, extraction methods affect not only yield but also the bioactivity of polysaccharides [54]. Therefore, it is pertinent to develop a suitable extraction method for better utilization of polysaccharides.

Thus, this study was carried out to explore the improved utilization of AKPs by investigating the effects of different extraction methods on AK. Currently, the main extraction methods for AK are HEW, UAE, and EAE. However, the current evaluation index only represents the yield [55]. Therefore, in this experiment, the AKPs were extracted according to the best parameters of three different techniques of HWE, UAE, and EAE respectively, and their synergistic activities were used for further evaluations. 

Three AKPs were extracted from AK, which had the same species of monosaccharide composition, all of which were dominated by arabinose, glucose, and galactose (unlike the polysaccharides obtained from AK previously by Qin et al. [15], which were dominated by fucose, arabinose and glucose). In addition, the different proportions of monosaccharides, molecular weight distribution, and glycosidic bonds of the three polysaccharides indicated the great influence of different extraction methods on the polysaccharides of AK.

At the same time, the synergistic effect of AKP on apatinib against SGC-7901 gastric cancer cells was determined for the first time. This is of great significance for the treatment of gastric cancers. SGC-7901 belongs to a common cell line of gastric cancer, which are related to the progress of malignant tumors [56]. There are many studies related to SGC-7901. For example, Wang et al. [57] obtained anthocyanins extracted from Chinese bayberry fruit which can significantly suppressed the growth of SGC-7901 in a dose-dependent manner. In this study, not only was the synergy of AKPs on apatinib against SGC-7901 determined, but also the association between the effect of AKPs and their properties was established. Synergistic effects were in the following decreasing order: UAE, HWE, and EAE. Compared with HAE-AKPs and EAE-AKPs, UAE-AKPs have the smallest molecular weight, β-configuration glycosidic bonds, and the highest galactose content. Low-molecular-weight polysaccharides are typically more water soluble [58] and can freely pass through biological membranes such as cell membranes and nuclear membranes [59]. Previous research found that the bioactivity of polysaccharides was strongly related to their galactose and galacturonic acid content [20,60]. Polysaccharides with β-configuration glycosidic bonds have better anti-tumor activity. [61]. Therefore, it can be assumed that these factors may contribute to the synergistic effect of UAE-AKPs.

## 5. Conclusions

The relationship between the physicochemical characteristics of AKPs extracted using three different methods and their synergistic activity were first investigated. According to the results, different extraction methods did not affect that make up the monosaccharide composition types of AKPs, although they did change the proportion of monosaccharides, the molecular weight, and the type of glycosidic bond. The results indicated that polysaccharides that contained a high galactose ratio, smaller molecular weight, and β-configuration exhibited better synergistic activity. Therefore, ultrasonic extraction could offer a good environment for the formation of this property or structure. Furthermore, the results showed that UAE was the simple and effective method of improving the synergistic activity of AKPs as a natural synergist for the treatment of gastric cancer.

## Figures and Tables

**Figure 1 molecules-27-04727-f001:**
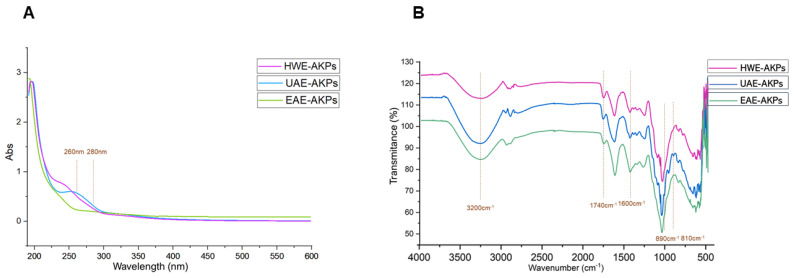
UV spectra(**A**) and FT-IR spectra (**B**) of three AKPs extracted by different extraction methods.

**Figure 2 molecules-27-04727-f002:**
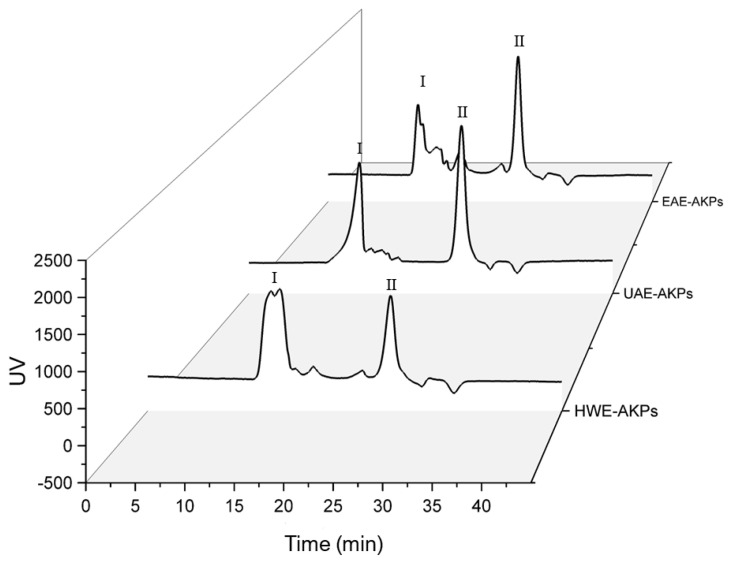
Molecular weight distribution of AKPs extracted by different extraction methods. There are two peaks, characterized as high-Mw parts (≥3100 kDa, defined as peak I), low-Mw parts (10–4 kDa, defined as peak II).

**Figure 3 molecules-27-04727-f003:**
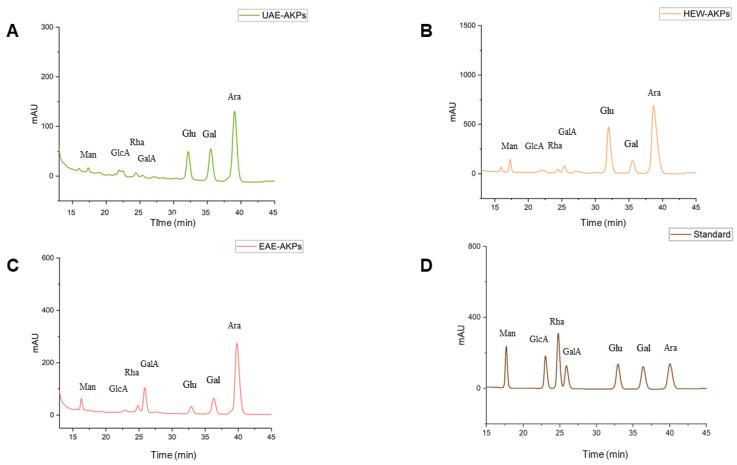
HPLC chromatogram for monosaccharide composition: UAE-AKPs (**A**), HWE-AKPs (**B**), EAE-AKPs (**C**) and standard monosaccharides (**D**). Mannose (Man); glucuronic acid (GlcA); rhamnose (Rha); galacturonic acid (GalA); glucose (Glu); galactose (Gal); arabinose (Ara).

**Figure 4 molecules-27-04727-f004:**
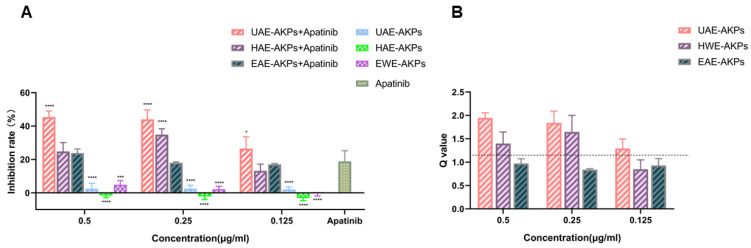
AKPs affected the anti-proliferate effect of apatinib on SGC-7901 cells. Inhibitory effect (**A**). Q value (**B**). The dashed line in the Figure 4B shows the Q value of 1.15. Data are representative as mean ± SEM. * *p* <0.05, *** *p* <0.001, **** *p* <0.0001.

**Figure 5 molecules-27-04727-f005:**
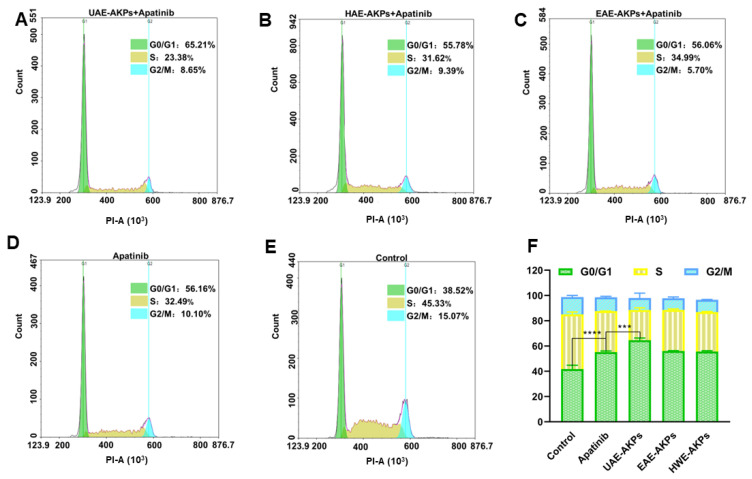
Effects of single or combined treatments on the cell cycle of SGC-7901 cells ((**A**) UAE-AKPs+ apatinib group; (**B**) HWE-AKPs+ apatinib group; (**C**) EAE-AKPs+ apatinib group; (**D**) apatinib group; (**E**) control group; (**F**) cell cycle). Data are representative as mean ± SEM. *** *p* <0.001, **** *p* <0.0001.

**Table 1 molecules-27-04727-t001:** Comparison of extraction quality, yield and total sugar of polysaccharides from *Atractylodes chinensis* (DC.) Koidz. (AKPs) extracted by ultrasound assisted extraction (UAE), hot water extraction (HWE) and enzymatic extraction.

Methods	UAE	HWE	EAE
Water to raw material ratio (mL/g)	20:1	20:1	20:1
extraction duration (min)	40	240	92
Extraction temperature (°C)	60	90	53
Ultrasonic Power (W)	160	/	/
Condition	PH = 7	PH = 7	1.2% complexenzyme; PH = 5
Extraction quality of polysaccharides (g)	3.27	13.44	2.125
Extraction rate of polysaccharides (%)	3.27	13.44	2.125
Total sugar of polysaccharides (%)	93.98	91.12	96.85

**Table 2 molecules-27-04727-t002:** The monosaccharide composition of AKPs extracted with three different methods.

Monosaccharide Composition (%)	UAE-AKPs	HWE-AKPs	EAE-AKPs
Mannose (Man)	0.17	4.12	0.11
Glucuronic acid (GlcA)	0.86	0.33	0.67
Rhamnose (Rha)	0.50	0.83	1.41
Galacturonic acid (GalA)	0.05	3.67	16.46
Glucose (Glu)	17.92	28.86	4.78
Galactose (Gal)	22.80	7.54	12.65
Arabinose (Ara)	57.70	54.65	64.02

## Data Availability

The data presented in this study are openly available.

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
