# Peer review of "A Comparison Study on Polysaccharides Extracted from Atractylodes chinensis (DC.) Koidz. Using Different Methods: Structural Characterization and Anti-SGC-7901 Effect of Combination with Apatinib"

_molecules, 2022, doi:10.3390/molecules27154727_

Round 1

Reviewer 1 Report

The manuscript molecules-1787832 devoted the actual field of medicinal chemistry, namely comparison study on polysaccharides and their effect of combination with apatinib and can be interested to the specialists working in this field. The authors’ opinion is clear and based on a good experimental material. The paper fit the Journal scope and formal requirements. However, it needs major revision before publication.

To improve the quality and perception of the manuscript I would suggest paying attention to following comments:

 1.      The rationale and strategy for the experiment design were not sufficient, and had critical flaws. The introduction should be reworked. The relationship between apatinib and polysaccharides is unclear. The authors must more clearly justify the feasibility of the research and their design.

2.      Given the specifics of the journal, it is necessary to explain the formulas of apatinib and the main components of polysaccharides.

3.      In table 1, the data should be given with the same accuracy (should be to 2 dp, but not to 3 dp).

4.      It would be good to broaden the conclusions in the context of a more detailed presentation of ways to resolve the problem.

5.      References list should be carefully checked and journal style policy should be strictly followed (doi, all authors, citation rule for books, monographs and articles, etc).

6.      There are some grammar and orthographical errors in the manuscript, which should be corrected

My decision is major revision.

Author Response

Dear reviewers:

Thanks for your professional reviews and constructive comments on our manuscript entitled " molecules-1787832; A comparison study on polysaccharides extracted from Atractylodes chinensis (DC.) Koidz. Using different methods: structural characterization and anti-SGC-7901 effect of combination with apatinib". It has a great help in improving the quality of the manuscript. We have studied these comments carefully and tried our best to revise and improve the manuscript. The details are outlined with red color in revised manuscript.

Yours sincerely, 

Pingfan Zhou

E-mail: [email protected].

Reviewer 2 Report

The paper of Zhoua, et al. “A comparison study on polysaccharides extracted from At- 2 ractylodes chinensis (DC.) Koidz. using different methods ...” aimed to investigation of Atractylodes chinensis polysaccharides and bioactivity. I would like to make some remarks that have arisen after reading the manuscript.

Line 48: Atractylodes chinensis (DC.) Koidz. is a synonim of Atractylodes lancea (Thunb.) DC. The Nomenclature should be corrected. 

Lines 58-60: "... a lack of investigations of Atractylodes chinensis (DC.) Koidz. on polysaccharides" includes data about three fructans (1) Int. J. Biol. Macromol. 2019;136:341–351; doi:10.1016/j.ijbiomac.2019.06.088; (2) BMC. Chem. 2019;13(1):92, doi: 10.1186/s13065-019-0605-8; (3)  Int. J. Biol. Macromol. 2016;82:765–771, doi: 10.1016/j.ijbiomac.2015.10.082; and rhamnogalacturonan Int. J. Biol. Macromol. 2019;136:341–351; doi:10.1016/j.ijbiomac.2019.06.088. 

Line 102:  "AK material (100g) was dissolved ..." - it mas mixed, not dissolved.

Line 103: "... using three different techniques (HWE, UAE, EAE) ..." - abbreviations need description.

Line 106: static or dynamic extraction type used?

Lines 104-114: it was completely unclear why these extraction conditions were chosen.

Line 116: "The supernatant was concentrated to 0.5 mg crude drug per mL ..." - how did you control the concentration of supernatant?

Line 132: solvent information need.

Lines 135-138: The description of FTIR won't take up much lines. It should be described in full.

Line 141: solvent information need.

Lines 153-163: standard curve information need.

Lines 164-186: standard curve information need.

Line 232: It is very strange that ultrasonic extraction was so much less effective than water extraction (more than 4 times). Water bath extraction is inextensive type of extraction unlike ultrasound. 

Table 1: the number of decimal places must be the same; Man should be mannose.

Section 3.2: The UV-Vis spectra of HWE and UAE demonstrated the sholulders at 250-270 regions suggest the presence of protein or/and phenolic components. How can you explain so high total sugar content?

Section 3.3: FTIR spectra of three polysaccharides were similar including carboxyl region (1750-1600 cm-1) but monosacchride data indicating the differences in uronic acid content - about 1% for UAE and >17% for EAE. That should be impacted on the FTIR data but it has not observed. You should explain that.

Section 3.4: I'm not clear, are the components I and II on Figure 2 the same polymers for all three samples or not? The correct decription need. 

Discussion. I can't find the comparative data the data obtained and the known informantion about Atractylodes lancea polysacchrides. Whithout these data it is difficult to estimate the novelity of your results.  

References. The section need to be corrected accordingly to Author guideline. 

Author Response

(The authors gave the same response as above.)

Reviewer 3 Report

The manuscript entitled, "A comparison study on polysaccharides extracted from Atractylodes chinensis (DC.) Koidz. Using different methods: structural characterization and anti-SGC-7901 effect of combination with apatinib" by Pingfan Zhou, Wanwan Xiao, Xiaoshuang Wang, Yayun Wu, Ruizhi Zhao and Yan Wang needs a lot of corrections.

My main complaints are as follows:

1. The botanical taxonomy after the species name is not italicized

2. The authors did not specify, what morphological part of the plant was used in their study

3. The authors should provide explanations of abbreviations. In Table 1 - what does "Man" mean?

4. Table 1 should contain the extraction conditions

5. Whether the extracts have been lyophilized, in addition, to yield, the mass of the extract should be reported

6. In part of the discussion there is no discussion or reference to other biological tests on the same cell line. This should be described in detail based on the literature review

7. The main text of the manuscript, including the reference list, requires many editorial corrections.

Author Response

(The authors gave the same response as above.)

Round 2

Reviewer 1 Report

The authors took into account the comments of the reviewer and significantly improved the manuscript. My decision is accept.

Author Response

Dear reviewers:

Thanks for your professional reviews and constructive comments on our manuscript entitled " molecules-1787832; A comparison study on polysaccharides extracted from Atractylodes chinensis (DC.) Koidz. Using different methods: structural characterization and anti-SGC-7901 effect of combination with apatinib". We appreciate you reviewing our manuscript at this unprecedented and challenging time. We wish you, your family and community well. Your careful review has helped make our research clearer and more comprehensive. And we have made further improvements to the manuscript. The details are outlined with red color in revised manuscript.

Yours sincerely,

Pingfan Zhou

E-mail: [email protected].

Reviewer 2 Report

The authors improved the manuscript acording to my comments and the final version of the paper looks good and publicable. Good luck. 

Author Response

(The authors gave the same response as above.)

Reviewer 3 Report

I can't understand how the authors corrected the mistakes in just many places. It's weak. Please read the entire manuscript again! Please compare the spelling on lines 3 and 62. There are more such shortcomings. Why there are no sugar abbreviations in Table 2, but they are not explained in Figure 3.
